# Combination of Small Extracellular Vesicle-Derived Annexin A2 Protein and mRNA as a Potential Predictive Biomarker for Chemotherapy Responsiveness in Aggressive Triple-Negative Breast Cancer

**DOI:** 10.3390/cancers15010212

**Published:** 2022-12-29

**Authors:** Priyanka P. Desai, Kalyani Narra, Johanna D. James, Harlan P. Jones, Amit K. Tripathi, Jamboor K. Vishwanatha

**Affiliations:** 1Department of Microbiology, Immunology and Genetics, University of North Texas Health Science Center, Fort Worth, Texas, TX 76107, USA; 2Department of Internal Medicine, John Peter Smith (JPS) Oncology Infusion Center, Fort Worth, Texas, TX 76104, USA; 3Biosample Repository Facility, Fox Chase Cancer Center, Philadelphia, PA 19111, USA

**Keywords:** triple-negative breast cancer, annexin A2, small extracellular vesicles, neo-adjuvant chemotherapy, biomarker

## Abstract

**Simple Summary:**

Triple-negative breast cancer (TNBC) is a highly aggressive disease. It is marked by a lack of estrogen receptor (ER), progesterone receptor (PR), and human epidermal growth factor receptor 2 (HER2). Thus, treating TNBC patients with targeted therapy and having a biomarker for therapy response has always been a challenge for medical field experts. TNBC patient tumors have high expression of Annexin A2 (AnxA2) protein and mRNA. Here in this study, we demonstrate that the AnxA2 protein and mRNA highly associate with small extracellular vesicles (sEVs) at the later stages of TNBC which is marked by a better treatment response in TNBC patients. Thus, we introduce sEV-derived AnxA2 protein and mRNA as a combined biomarker for neo-adjuvant chemotherapy response in TNBC.

**Abstract:**

Small extracellular vesicles (sEVs), mainly exosomes, are nanovesicles that shed from the membrane as intraluminal vesicles of the multivesicular bodies, serve as vehicles that carry cargo influential in modulating the tumor microenvironment for the multi-step process of cancer metastasis. Annexin A2 (AnxA2), a calcium(Ca^2+^)-dependent phospholipid-binding protein, is among sEV cargoes. sEV-derived AnxA2 (sEV-AnxA2) protein is involved in the process of metastasis in triple-negative breast cancer (TNBC). The objective of the current study is to determine whether sEV-AnxA2 protein and/or mRNA could be a useful biomarkers to predict the responsiveness of chemotherapy in TNBC. Removal of Immunoglobulin G (IgG) from the serum as well as using the System Bioscience’s ExoQuick Ultra kit resulted in efficient sEV isolation and detection of sEV-AnxA2 protein and mRNA compared to the ultracentrifugation method. The standardized method was applied to the twenty TNBC patient sera for sEV isolation. High levels of sEV-AnxA2 protein and/or mRNA were associated with stage 3 and above in TNBC. Four patients who responded to neoadjuvant chemotherapy had high expression of AnxA2 protein and/or mRNA in sEVs, while other four who did not respond to chemotherapy had low levels of AnxA2 protein and mRNA in sEVs. Our data suggest that the sEV-AnxA2 protein and mRNA could be a combined predictive biomarker for responsiveness to chemotherapy in aggressive TNBC.

## 1. Introduction

Triple-negative breast cancer (TNBC) represents a lack of estrogen receptor (ER), progesterone receptor (PR), and human epidermal growth factor receptor 2 (HER2) [1,2]. TNBC constitutes 15–20% of all breast cancer cases and there is a high prevalence of TNBC in young black women [3]. TNBC is highly invasive with an increased likelihood of death with a dismal 13.3 months after the occurrence of distant metastases [4]. The major metastatic sites for TNBC cells are the lung, liver, bone, and brain [5]. American Joint Commission of Cancer (AJCC) recommends that oncologists use clinical prognostic staging, which incorporates biological markers and grades in addition to traditional TNM (tumor, node, distant metastases) anatomical staging. Upon diagnosis, breast cancer patients are assigned stages 1 to 4 with sub-classification of A or B. The lower stages of classification show better survival rates, and A subclass is more favorable than B [6]. Definitive surgery by itself is not curative for most patients with TNBC. Neoadjuvant systemic therapy is preferred for all patients in stages 2 and 3. Achievement of pathologic complete response (pCR), which is defined as having no invasive cancer visible at the time of surgery following neoadjuvant therapy, is associated with long-term survival outcomes in TNBC [7]. When patients do not achieve pCR, significant downstaging with lower residual cancer burden following neoadjuvant treatment is associated with better outcomes [8]. Stage 4 patients are generally treated with only systemic therapy. There is no current strategy to predict who is going to respond and all patients are offered the same treatment. For patients who have a low chance of response, the current approach will be inadequate and novel treatment regimens need to be developed [9].

There is an unmet need to develop a better regimen and biomarker for TNBC as well as other cancers. Small extracellular vesicles (sEVs) like exosomes, the nano-sized particles with a size range of 30–150 nm from the family of extracellular vesicles, are studied extensively in the field of cancer [10]. sEVs consist of nucleic acids, proteins, lipids, and metabolites. Some of the proteins in sEVs are tetraspanins Cluster of differentiation 9 (CD9), CD63, CD81, Tumor susceptibility gene 101 (Tsg101), Heat shock proteins (Hsp) like Hsp70, Hsp90, Rab proteins, Annexins, and cofilin which could also be used as an EV markers [11,12]. sEVs also contain a pool of double-stranded DNA, RNAs like mRNA, miRNAs, long non-coding RNAs (lncRNAs), tRNAs, etc. [13]. Tumor-derived sEVs can modulate the tumor microenvironment by interacting with the neighboring cells for the process of metastasis. The uptake of sEVs by the non-cancerous cells in the body fosters the pathological state of the normal cells [14,15]. Different sizes of EVs are known to have varied cargo content and shedding capacity from the cells. Alix, a sEV biogenesis marker, was found to be enriched in sEVs compared to big extracellular vesicles (bEVs).bEVs. These EVs have the potential to modulate organ tropism, in TNBC. sEVs were predominantly detected in the liver, whereas, bEVs remained detectable in the liver, kidney, lung, and spleen. This shows that sEVs and bEVs are distinct EV populations which could cause tumor progression with differential biodistribution properties [16] As EV size is an important determinant [17,18], we have compared the sizes of sEVs derived from different stages of TNBC patient sera in our study.

Recently, sEV biogenesis and release have been extensively studied for cancer therapy and diagnosis. sEVs have emerged as a diagnostic tool in cancer detection. In 2016, Hannafon et al. showed that plasma sEV-derived miRNA-21 and miRNA-1246 could be combined for better prognosis for breast cancer patients [19]. In lung cancer where invasive methods like transbronchial needle aspiration or transthoracic biopsy are used as the diagnosis methods, sEV markers like CD151, CD171, and tetraspanin 8 were identified as a combined diagnostic marker in 276 Non-small cell lung carcinoma (NSCLC) patients [20]. Urinary sEV-derived miR-204-5p could be used for an early diagnosis of RCC [21]. Yang et al. showed that circular RNA, like circPSMA1 from sEVs, could be an independent prognostic marker for diagnosis and therapy of TNBC [22].

It’s been known that chemotherapy treatment induces EV release and modulation in cargo content, so it’s important to study the chemotherapy drugs and its surrounding changes in the patient body [23]. Drugs like carboplatin, paclitaxel, and irinotecan could elevate the exosome release in hepatocellular carcinoma [24]. The number of exosomes shedding was also seen to be increased when MDA-MB-231 cells were treated with paclitaxel compared to control cells [25]. There was another study which have demonstrated some discrepancies like no significant increase in numbers of EVs have been observed after treatment of ovarian cancer cells with cisplatin [26]. Ludwig et al. found that there was reduction in exosomal protein after oncological therapies in head and neck cancer [27]. Chemotherapies also regulate the loading of messenger RNA (mRNA), microRNA (miRNA), and non-coding RNA. A recent study by Shen et al. in 2019, showed that EV-released miRNA by chemotherapy induction leads to breast cancer stemness [28].

As EV cargo loading and release is essential to study the aggressiveness of the disease, we have measured the number of sEVs and protein content released by the TNBC patient in different stages.

Annexins are a family of Ca^2+^-dependent phospholipid-binding proteins. There are 12 human annexins in the family with diverse functions like blood coagulation, inflammation, migration, invasion, angiogenesis, etc. [29,30]. The study supporting that the high Annexin A3 expressing ovarian cells release more sEVs, also demonstrated that the cells when are cisplatin-resistant load more Annexin A3 into the sEVs [31]. Furthermore, Keklikoglou and team showed that breast cancer patients undergoing neoadjuvant chemotherapy have higher expression of Annexin A6 in EVs compared to pre-treatment levels [32].

So, here we introduce AnxA2, as a Ca^2+^-dependent phospholipid-binding protein, and an AnxA2 mRNA-binding protein [33] which has been implicated in many cancers like breast, prostate, ovarian, etc., as it helps cancer cells to migrate, invade, and proliferate in the tumor microenvironment [34]. AnxA2 has also been characterized as a biomarker in HER2 and luminal B breast cancer [35]. It is been previously observed that the sEV-AnxA2 protein creates a pre-metastatic niche in TNBC via macrophage-mediated activation of Mitogen-associated protein kinase (p-38MAPK), Nuclear Factor Kappa-light-chain-enhancer of activated B-cells (NF-κB), and Signal transducer and activator of transcription (STAT3) pathways. This shows the important role of the sEV-AnxA2 protein in breast cancer metastasis [36]. We have also shown that serum sEV-AnxA2 protein is high in breast cancer patients and is associated with tumor grade, poor overall survival, and disease-free survival [37].

However, we have yet to explore the expression of sEV-AnxA2 protein and its mRNA in different stages of TNBC and whether the combination of both protein and mRNA could associate with therapy response in the patients. Thus, in this study, we seek to determine the expression of sEV-AnxA2 protein and sEV-AnxA2 mRNA in different stages of TNBC patients. We first time propose to use them as a combined biomarker in the higher stages of TNBC for predictive chemotherapy response for the treatment of TNBC patients.

## 2. Results

### 2.1. Stages of Cancer in TNBC Patients Did Not Show a Significant Effect on the Size and the Release of the sEVs

To achieve the highest purity of vesicles with the least expression of contaminants from other organelles, albumin and IgG, we standardized the sEV isolation protocol on mouse serum (Figure 1). The samples were pre-treated by depleting IgG from the mouse serum before subjecting the samples to the ExoQuick Ultra kit. The sEV sizes were determined by NTA analysis (Appendix A). Minimal contaminants were observed in sEVs isolated by the ExoQuick Ultra kit after IgG depletion (Appendix A) compared to significant expression without IgG depletion (Appendix A).

We isolated sEVs from the normal and TNBC patient sera. We selected 17 patient samples from a set of over 80 samples and calculated their clinical prognostic stages. The sizes of the sEVs in all the patient samples were in the range of 86–106 nm (Figure 2(Ai–Aiv) and Appendix A). Cryo-transmission electron microscopy visualization showed that the vesicles were in the size range of 20–80 nm. The sEVs were isolated from the stage 3B TNBC patient serum (Patient no.-14) showing a mixed population of double-membraned exosomes with sizes ranging from 50–80 nm, and non-membranous vesicles called exomeres as mentioned by Zhang et al. 2018, with a size range of 20–40 nm (Figure 2B) [38].

We decided to compare the sEVs release and their respective sizes derived from TNBC patients of different stages as it is known that the vesicle size could lead to tumor progression via activation of different pathways by releasing the EVs [39,40]. Our data showed that the sEVs sizes remained unchanged irrespective of the TNBC stages 1, 2, 3 and above. The mean size of sEVs also did not show any significant change when compared between the TNBC stages (Figure 2C). The sizes of the sEVs from all the TNBC patients are given in Appendix A. The NTA analysis of the sEVs isolated from the individual patient serum samples showed no trend of sEVs release in the patients with different clinical prognostic stages. When means of the number of sEVs released per ml were compared between the stages of TNBC, no significant change was found between stages 1, 2, 3, and above (Figure 2D).

### 2.2. Elevated Expression of AnxA2 Protein in sEVs Is Associated with Advanced TNBC Stage and Improved Response to Neoadjuvant Chemotherapy

The AnxA2 protein expression in sEVs was observed to be minimal in the normal serum-derived sEVs as well as samples with stage 1B (lanes 1,2,3,4,10,11,12,19,20,21 in Figure 3A,B). The patient number 7 was considered a standard sample to compare with other samples from higher stages because the patient was suffering from stage 1 and did not undergo chemotherapy. The patients with stage 2 showed lower expression of AnxA2 protein in sEVs compared to stage 3 and above (lanes 5,6,7,8,13,14,15,16,17,22,23,24,25,26 in Figure 3A,B) and higher than stage 1 (lanes 3,4,5,6,11,13,20,23 in Figure 3A,B). Out of 9 patients from stage 3 and above, 4 patients (lanes 8,15,16 and 25 in Figure 3A) showed high AnxA2 protein expression in sEVs, i.e., 44.44% of stage 3 and above patients (Figure 3A,B). The relative fold change was calculated by considering all the sEV samples including normal and all the TNBC stages. The fold change in sEV-derived AnxA2 protein expression was observed to be significantly higher in stage 3 and above compared to normal and stage 1 TNBC samples. There was no significant change in the sEV-derived AnxA2 protein expression between normal and stage 1, normal and stage 2, and stage 2 and stage 3 and above TNBC serum-derived sEVs (Figure 3C). There was no detectable expression of calnexin in any of the patient sample-derived sEVs. However, four normal/TNBC serum sEV samples showed an expression of IgG and albumin (lanes 1,4,7, and 19 in Figure 3A; normal serum samples no. 2, and 3; patient no.-7, and 12). The remaining sEV samples showed minimal expression of both proteins.

As patient sample numbers 1,3,4,5,6,9, and 12, showed the least expression of sEV markers (Figure 3A), we loaded more sEVs (1.5 × 10^9^) to check the presence of the sEV protein markers in these samples. Out of 7 patients, patients numbers 1,3, and 12 showed significant expression of AnxA2, CD81, CD9, and Tsg101. Patients 4 and 5 showed AnxA2 and CD9, whereas patient 6 showed CD9 and Tsg101 sEV protein expression. Patient 9 showed only one sEV marker, CD9 (Appendix A). As we could observe less expression of sEV markers in patients 6 and 9, we carried out a Western blot by loading more sEVs (2 × 10^9^). Both the patients showed significant expression of AnxA2 and CD9 protein. Tsg101 protein expression was observed to be significant in patient 6, whereas patient 9 had very low expression of the protein (Appendix A). Taking all the sEV patient-derived samples into account, we observed AnxA2 and CD9 as sEV markers present in maximum samples (Figure 3A and Appendix A).

The chemotherapy administration records revealed that there was a correlation between the responsiveness to neo-adjuvant chemotherapy and AnxA2 protein expression in sEVs derived from patients with stage 3 and above. Out of 8 patients with stage 3 and above who received neoadjuvant chemotherapy, 5 patients had low expression of AnxA2 protein in sEVs and only 1 (20%) had a partial response. All 3 (100%) patients with high AnxA2 protein expression in sEVs responded well with significant downstaging of the tumor to neoadjuvant chemotherapy (Table 1). The ROC curve was constructed to validate the sEV-derived AnxA2 protein as a biomarker for effective response to chemotherapy. The AUC was seen to be 0.875 (95% Cl: 0.6938–1.056; *p*-value = 0.011) predicting it as a good indicator of chemotherapy response in stage 3 and above TNBC (Figure 3D). As per the ROC curve calculations, the highest sensitivity, specificity, and accuracy to predict the chemotherapy response with a cut-off value of 2.95 fold increase for sEVs derived AnxA2 protein expression as a biomarker alone were 75%, 100%, and 87.5%, respectively (Table 2).

**Table 1 cancers-15-00212-t001:** The expression of the AnxA2 protein and mRNA derived from sEVs in the TNBC patients with stage 3 and above.

Patient No.	Clinical Prognostic Stages	Stages after Chemotherapy	Response to Chemotherapy (Yes/No)	Expression of AnxA2 Protein in sEVs	Expression of AnxA2 mRNA in sEVs
12	3B	3C	No	Low	Negligible
13	3B	2B	No	Low	Low
18	3C	3C	No	Low	Low
19	3C	3C	No	Low	Low
15	3B	1B	Yes	High	High
16	3B	1A	Yes	Low	High
17	3B	1A	Yes	High	High
20	4	0	Yes	High	High

**Table 2 cancers-15-00212-t002:** Details of the cut-offs, sensitivity, specificity, accuracy, Youden’s Index, Area Under Curve values, and *p*-values of AnxA2 protein or mRNA derived from sEVs as a biomarker alone or in combination for neoadjuvant chemotherapy response in TNBC.

Parameters	Cut-off (Fold Change)	Sensitivity (%)	Specificity (%)	Accuracy (%)	Youden Index	AUC (95% CI)	*p*-Value
Expression of AnxA2 protein in sEVs	2.95	75	100	87.5	0.75	0.875	0.012
Expression of AnxA2 mRNA in sEVs	2.33	100	100	100	1	0.844	0.021
Combination of expression of AnxA2 protein + AnxA2 mRNA in sEVs	2.33	87.5	100	93.75	0.875	0.891	0.008

**Figure 3 cancers-15-00212-f003:**
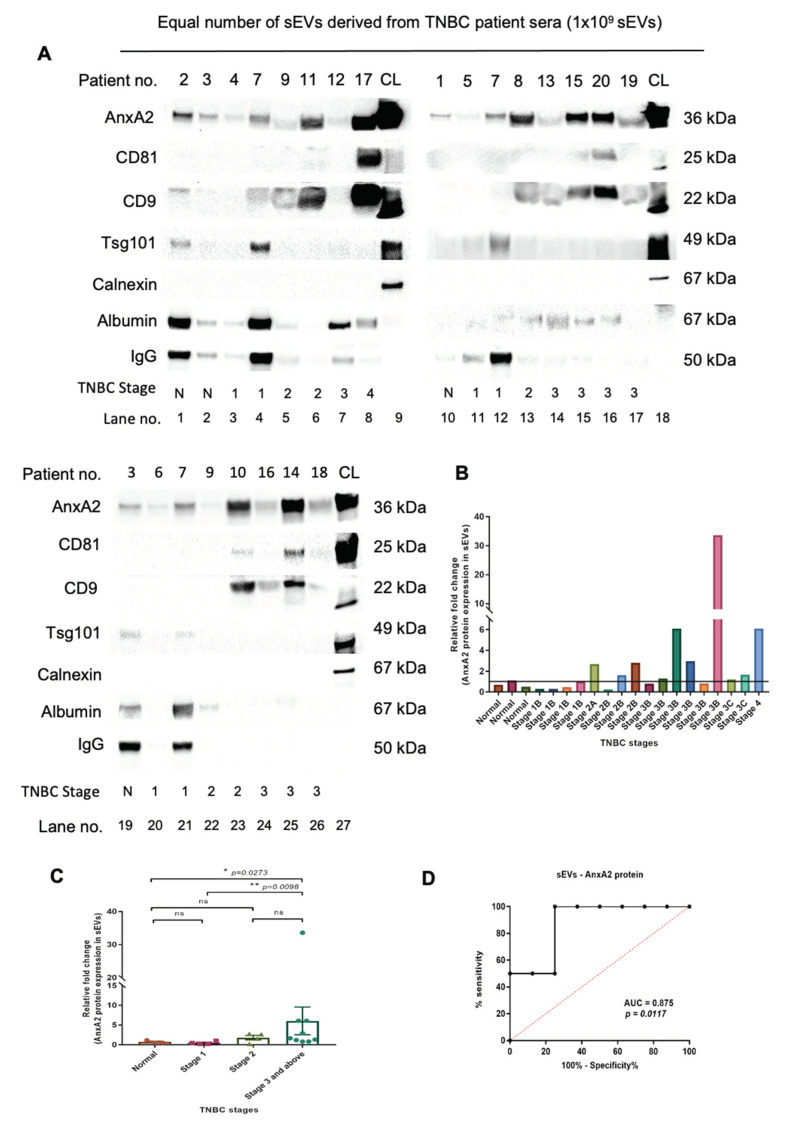
Later stages of TNBC are marked with increased expression of the sEV-derived AnxA2 protein in TNBC patients and it is significantly associated with responsiveness to the chemotherapy. (**A**) Western blot showing AnxA2 protein expression in the equal number of sEV-derived from TNBC patient serum according to the stages of cancer, CL, cell lysate. The patient information corresponds to the patient number as described in Table 3 (normal samples—1,2,3; the patient numbers with stage 1-no chemotherapy-4,5,6,7; stage 2-no chemotherapy-8,10, and 11; stage 2 with chemotherapy—9; stage 3 and above-no chemotherapy-14; stage 3 and above with chemotherapy-12,13,15,16,17,18,19,20). (**B**) The graphical representation of the increased sEV-AnxA2 protein expression in the fold change compared to the stage 1 samples. The patients are as described in Table 3 from 1–20. (**C**) The graphical representation of the mean fold change of the AnxA2 protein expression in the sEV-derived from the TNBC patients from each stage. (mean ± SEM, n = 3 for normal samples, n = 4 for stage 1, n = 4 for stage 2, and n = 9 for stage 3 and above; ** *p* < 0.01, * *p* < 0.05). (**D**) ROC curve for serum sEV-derived AnxA2 protein as a biomarker alone for the prediction of responsiveness or non-responsiveness to neoadjuvant chemotherapy in patients with stage 3 and above (n = 8).

### 2.3. Elevated Expression of AnxA2 mRNA in sEVs Is Associated with Higher TNBC Stages and Response to Neoadjuvant Chemotherapy

We subjected the 20 sEV samples for RNA isolation and a qRT-PCR. We considered all the values below 1 as a negligible expression of mRNA in the samples. With the above settings, we observed very low expression of sEV-AnxA2 mRNA in the normal and samples with stage 1B serum-derived sEVs as shown in the first seven samples (Figure 4A). In four, stage 2A and 2B, patients, we did not observe any significant change in the sEV-AnxA2 mRNA expression compared to stage 1B (Figure 4A). Out of 9 patient samples of stage 3 and above, 4 showed high AnxA2 mRNA expression in sEVs which is 44.44% of all the stage 3 and above patients (Figure 4A).

The mRNA analysis of sEV samples showed that there was a significant increase in sEV-AnxA2 mRNA expression in stages 2 and 3 and above-derived sEVs compared to the normal serum-derived sEVs. There was no significant change in the expression of AnxA2 protein when compared between stages 1, 2, and stage 3 and above (Figure 4B).

Of the patients diagnosed with stage 3 and above, 4 out of 8 patients were non-responsive to the neoadjuvant chemotherapy and had low expression of sEV-AnxA2 mRNA. The other 4 (100%) with high sEV-derived AnxA2 mRNA expression responded well to the chemotherapy with significant downstaging of the tumor. Thus, we constructed the ROC curve with sensitivity and specificity values of each score. ROC curve showed the AUC of 0.8438 (95% Cl: 0.6467–1.041; *p* = 0.021), predicting that sEVs derived AnxA2 mRNA could be a good predictor of neoadjuvant chemotherapy response in the TNBC patients with the cut-off value of 2.33 fold increase with the highest specificity, sensitivity, and accuracy of 100%, 100%, and 100%, respectively, to use sEV-AnxA2 mRNA levels as a biomarker alone (Figure 4C and Table 2).

The data indicate that AnxA2 protein derived from sEVs could potentially be a better predictive marker for chemotherapy responses in stage 3 and above with an AUC of 0.875 compared to sEV-derived AnxA2 mRNA as discussed above.

### 2.4. AnxA2 Protein and mRNA Derived from sEVs as a Combined Biomarker for Prediction of the Effectiveness of Chemotherapy in the TNBC Patients

Of 20 tested samples, 14 samples had lower levels of sEV-derived AnxA2 protein and mRNA, and 3 had high protein and mRNA expression. This shows an 85% concordance rate between protein and mRNA expression. Only 3 samples showed differences in the levels of protein and mRNA, with patient no. 16 having low protein and high mRNA whereas patients 8 and 14 had high protein and low mRNA expression (Figure 3B and Figure 4A, Table 3).

Because of the high concordance between sEV-derived AnxA2 protein and mRNA and as testing for both protein and mRNA could increase the likelihood of finding the sEV-AnxA2 expression, we decided to construct the combined ROC curve by using logistic regression. The combined ROC curve showed an AUC of 0.8906 (95% Cl: 0.6863–1.000; *p*-value = 0.0087). The likelihood ratio test for the combined marker showed a log-likelihood ratio of 12.57 and a *p*-value of 0.0004. We observed that the AUC value obtained from the combined ROC curve of protein and mRNA alone was higher compared to the AUC values obtained from protein and mRNA (AUC = 0.875 and 0.843, respectively). The sensitivity and accuracy were observed to be improved with combination markers, 87.5%, and 93.75%, compared to protein alone, 75% and 87.5%, respectively, with a cut-off value of 2.33 fold increase. This indicated that the combination marker could be a good predictor of chemotherapy response in TNBC patients with stage 3 and above (Figure 5 and Table 2).

## 3. Discussion

Our study aims to establish sEV-AnxA2 as a biomarker and identify the inter-relationship between its levels in various stages of cancer progression in TNBC patients. In the 21st century, the medical field has witnessed tremendous advances in the field of basic and clinical cancer research. Even still, the diagnosis and treatment of cancer is a common challenge for clinicians and researchers in the field [41]. Breast cancer is the leading cause of global cancer incidence, according to the 2020 World Health Organization (WHO) report, having surpassed lung cancer. Among women globally, breast cancer accounts for 1 in 4 cancer cases and for 1 in 6 cancer deaths [42]. One in eight American women have a risk of developing breast cancer in their lifetime [43,44].

Several molecular predictive markers have been studied in TNBC but none have made it into the clinics. So, there is an urgent need for a biomarker to predict the responsiveness of neoadjuvant chemotherapy for the treatment of TNBC patients. This need for a predictive marker can inform clinicians of the chance of a patient’s response to a particular systemic therapy so that a treatment plan can be modified for better therapy outcomes [45]. Hence, we intend to use the combined sEV-derived AnxA2 protein and mRNA biomarker to predict the patient tumor stages, grades, and to study the best suitable drug for the early treatment in TNBC patients. Also, we would carry out the study in different ethnic groups to predict the chemotherapy response in TNBC.

We studied the sEVs secreted by tumors in response to the aggressiveness of the disease. Their functions as delivery vehicles of biological material make sEVs promising biomarkers for the early prediction of disease progression and drug resistance in breast cancer, as well as for therapeutic targeting of molecules to treat this deadly disease [46]. Considering the multifunctional activities of sEVs in pathological conditions, there is a need to develop a diagnostic tool using these messenger guides, or a therapeutic approach, to minimize the protein loading into the sEVs or their secretion in pathological conditions. Cancer sEVs are a good tool in many cancers like prostate, breast, ovarian, and glioblastoma to treat or diagnose the disease [47].

sEVs have been an emerging candidate compared to circulating tumor cells (CTCs) or circulating tumor DNA (ctDNA) in studying or treating the disease [48]. The main reason to use the sEVs for liquid biopsy is a result of their inherent stability due to their lipid bilayer. They are stable at 4 °C for a week and could then be stored at −80 °C for months [49]. Secondly, sEVs represent a replica of their parental cell as they carry numerous biological information like proteins and nucleic acids in them. The information we received after isolating sEVs represents the diseased state of the cell [50,51]. Thirdly, now we have a technique to isolate sEVs from 200 µL of patient serum as compared to the isolation of CTCs from 1 mL of blood. CTCs are few in number, even in a large quantity of serum samples, and could tend to miss the information because of a lack of a technique to isolate the CTCs from the patient samples [50].

A small study previously suggested that AnxA2 protein positivity in tissue specimens correlated with poor pathologic response [52]. Exo-Carta database shows AnxA2 is the protein found in abundance in the sEVs [53]. sEV-derived AnxA2 protein is involved in transporting sEVs out of the TNBC cells with the potential to get transferred from one cell to another [54]. sEV-AnxA2 protein derived from ovarian cancer cells is shown to induce epithelial-mesenchymal plasticity in human peritoneal mesothelial cells. sEV-AnxA2 protein could promote invasion, migration, morphological changes, and fibrosis in normal peritoneal mesothelial HMrSV5 cells. Also, sEV-AnxA2 protein leads to mesothelial-mesenchymal transition by degradation of the extracellular matrix in the tumor microenvironment to create a pre-metastatic niche for ovarian cancer development through the PI3K/Akt/mTOR pathway [34]. Thus, studying sEV-AnxA2 protein as a target or as a diagnostic tool in different stages of cancer is very essential.

Our previous analysis of the Cancer Genome Atlas (TCGA) cohort of 113 TNBC patients showed a poor survival rate due to high AnxA2 tissue mRNA expression compared to ER-positive, PR- positive, and HER2-positive breast cancer patients [55]. According to a previous study, the AnxA2 protein binds to its cognate mRNA at approximately 100 nucleotides spanning the region of the 3′ untranslated region (UTR) [56]. Thus, with the previous research showing the high expression of AnxA2 mRNA in TNBC patients and binding of AnxA2 protein and mRNA, we investigated the association of sEV-AnxA2 mRNA in the patients for the first time. A significant amount of AnxA2 protein and mRNA in the patient serum-derived sEVs directed us to question whether the sEV-AnxA2 protein and mRNA have an impact on aggressiveness in TNBC patients, which could lead to failures of the treatments.

When studying the sEV-AnxA2 protein and mRNA, we worked on standardizing the sEV isolation protocol and considered the guidelines mentioned in MISEV 2018 during the study [57]. As AnxA2 is a calcium-dependent phospholipid binding [58] and an mRNA binding protein that associates with the cell surface and is present inside the sEVs, we decided to standardize the protocol for the least loss of sEV surface AnxA2 protein or mRNA. To improve the purity and yield of sEVs to detect the sEV-AnxA2 protein and mRNA in a low amount of serum sample, we ensured that the protocol had steps to help detect the sEVs and the other organelles marker protein with AnxA2 protein in the Western blot technique. We used the C57BL/6 mice serum samples for standardization of the protocol to investigate the reliability of the protocol in samples derived from two different species: mice and humans. Ultracentrifugation, which is a gold standard method [59], had the disadvantage of less yield from the 200 μL mouse serum sample. We could not detect sEV membrane proteins like CD9 and AnxA2. Also, we had GM130, Golgi body protein, after IgG removal, which showed that the eluted sample was not pure sEVs but mixed with contaminants. The major problem related to the patient serum-derived sEVs has been the downstream application due to the high amount of IgG, lipid, or albumin present in the pelleted sEVs [60]. Hence, we diluted the mice serum with PBS and depleted the maximum pool of IgG to make the sample less viscous in order to pass through all the steps without a major loss of proteins. Finally, samples were subjected to the ExoQuick Ultra kit. ExoQuick Ultra kit is known to yield a high number of sEVs with less purity [61]. However, the combination of IgG depletion and the kit helped us gain more purity with few traces of calnexin and GM130 in the mice serum-derived sEV samples.

Thus, after confirming the significant yield and purity by NTA and Western blot, we used the protocol on the TNBC patient sera to detect the sEV-AnxA2 protein and mRNA. The majority of TNBC serum-derived sEV samples (16 out of 20) had very faint bands of IgG and albumin which shows that the protocol subjected was able to isolate the sEVs with a minimum amount of contaminants. This combination helped us properly separate the samples on the SDS-PAGE and the protein expression in human serum-derived sEV samples in the Western blot. We confirmed that with the standardized protocol, we had a negligible expression of calnexin in all the human TNBC sample-derived sEV pellets.

But, here we would also like to highlight that handling the patient-derived sEVs and their storage is crucial because mishandling samples by repeated freeze-thawing or storage conditions could hamper the downstream experimental results [62,63]. Though −80 °C prevents changes in the physical and functional properties of sEVs, it has been shown that sEVs could lose miRNA over 90 days period at −80 °C. Also, sEVs storage at −20 °C and 4 °C leads to a decrease in miRNA levels upto 50% in 30 days. Storage at −20 °C did not change size but sizes of sEVs continue to decrease over 90 days at 4 °C [64]. Also, one of the studies illustrated that a single freezing cycle at −20 °C, could lead to a loss of 10–15% of sEVs [65], on the other side, another study demonstrated that 1–10 freezing cycles do not affect sEV sizes [66].

Previously, we showed that the sEV-AnxA2 protein could be a good prognostic marker in TNBC via the tumor grading [37]. Here, we extended the study and proposed that both sEV-AnxA2 protein and mRNA could be used in combination to ensure a better prognostic marker in higher stages and a biomarker for chemotherapy response in TNBC. Following this, we worked on the 17 TNBC patient sera-derived sEVs to analyze the levels of sEV-AnxA2 protein and mRNA in TNBC patients in different stages. We would like to highlight that this study could significantly contribute to the EV and the cancer therapy field because of the high concordance of 85% between sEV-AnxA2 protein and mRNA in TNBC patients. We also observed that out of 9 patients with stage 3 and above, one of the patients (Patient 16) had less sEV-AnxA2 protein and more sEV-AnxA2 mRNA. We believed that AnxA2 mRNA could associate with vesicles via its protein binding, but we yet have to confirm at the experimental levels. The discrepancy between sEV-AnxA2 protein and mRNA results could be an outcome of an impaired AnxA2 mRNA translation mechanism in the patient. Also, AnxA2 mRNA could have a different mechanism to associate with the vesicles compared to protein. The major pathway could be via the autophagosomal pathway as AnxA2 is also an autophagic protein. However, this is the subject for further research to study the sEV-AnxA2 mRNA mechanism for its loading into the sEV, whether loading is AnxA2 protein-dependent or independent. These are the few questions we would like to pursue to improve our knowledge in the field of AnxA2 and extracellular vesicles [67,68].

Additionally, we would like to highlight our Cryo-TEM data where we observed the different sizes of vesicles ranging from 20–80 nm in the stage 3 TNBC patient serum. The population had few non-membranous exomeres and double-membrane exosomes. The proteomic profiling of the sEVs has shown the high expression of Annexins and CD9/63/81 proteins (tetraspanins) compared to exomeres [38]. This could be one of the possibilities that the patient 9 serum-derived sEVs had a high number of exomeres compared to sEVs which resulted in less expression of sEV markers. Though we cannot assure the exact reason behind the discrepancy in the EV markers, there could be a few more factors like handling of the blood, time of blood collection, haemolysis, or removal of platelets [69]. The residual platelets if present in serum could contribute via platelet-derived EVs. Platelet-derived EVs are said to be more positive for CD9 protein [70]. Patient 9 also had a very minimal expression of one of the EV markers that is CD9. Also, the heterogeneity in TNBC leads to the release of a wide range of extracellular vesicles of varying sizes. Hence, we need a better technique to isolate the extracellular vesicles to study their protein cargoes and biogenesis for the best treatment or diagnostic approach for cancer patients [71,72,73].

One of the drawbacks of our study is that it is a retrospective study of samples collected at different time points during the patient’s treatment regimen. We are aware that the study encompasses a small sample size and hence, extensive studies are needed to validate the sEV-AnxA2 as a chemotherapy response biomarker before it gets translated to clinical settings. If a large number of samples show significant differences in treatment responses with improved sensitivity, specificity, and accuracy, then sEV-AnxA2 protein and mRNA could help increase the predictive value of chemotherapy response in advanced TNBC and for a possible clinical decision. Also, future studies need to be designed to help validate our findings by collecting sEV-AnxA2 protein and mRNA at various time points serially in more patients at diagnosis and before starting neoadjuvant chemotherapy, after the completion of chemotherapy, and before definitive surgery. If we can establish a predictive biomarker, we can help tailor treatment approaches to patients armed with this knowledge. This can help deliver effective treatments that have a higher probability of response and minimize unnecessary side effect burdens. Also, the biology behind the effect of chemotherapy on the sEV in TNBC patients needs to be evaluated in terms of cargo loading and sEV release in tumor microenvironment. The study also impose the question for future study that how the increased expression of AnxA2 protein and mRNA in sEVs could show better response to chemotherapy compared to sEVs bearing low AnxA2 protein and mRNA in the current patient cohort.

So collectively, by considering all the limitations and drawbacks of the study, we propose that the sEV-AnxA2 protein and sEV-AnxA2 mRNA could be explored as a predictive combined biomarker in TNBC patients for better chemotherapy outcomes.

## 4. Materials and Methods

### 4.1. Animals and Blood Collection

Animal work was approved by Institutional Animal Care and Use Committee (IACUC). 8 months old C57BL/6 mice were purchased from Taconic Biosciences. All the procedures were carried out according to the IACUC rules and regulations. The mice were housed in plastic cages at a controlled temperature of 22 ± 1 °C on a 12 h light/12 h dark cycle with lights on from 0600 to 1800 h. Standard rodent chow and water were provided throughout the experimental study. Blood was collected by retroorbital bleeding. All the procedures were carried out under Isoflurane anesthesia.

### 4.2. Serum Collection from Mouse Blood

After collecting the blood, it was allowed to clot for 45 min at 37 °C. The tubes were placed overnight at 4 °C to allow the clot to contract. The tubes were centrifuged at 400× *g* for 10 min at 4 °C to remove any remaining insoluble material. The supernatant was transferred to the new tube and centrifuged at 3000× *g* for 10 min at 4 °C to pellet down the debris.

### 4.3. Human TNBC Serum Samples

The serum samples of triple-negative breast cancer patients were procured from the Fox Chase Cancer Center. The blood samples were collected in a Red-top tube and then processed for serum collection. After collection, the samples were stored at −80 °C. All serum samples were purchased and used under the Institutional Review Board (IRB)-approved protocols at the site of collection and the University of North Texas Health Science Center at Fort Worth in Texas. These samples were drawn between the years 2011 to 2017. Information collected included clinical T, N, and M stages at diagnosis of TNBC and the grade of the tumor. Using this information, we performed stratified random sampling. We divided patients according to their TNBC stages. The clinical prognostic stage was calculated using AJCC version 8. We further selected samples randomly from each stage group. Additional information obtained included the year of the sample drawn, whether patients had definitive surgery for treatment, and whether neoadjuvant chemotherapy was given prior to surgery. An interval from the start of chemotherapy to surgery was obtained for patients who were given neoadjuvant chemotherapy. Finally, information was obtained on an interval from a sample drawn relative to the date of surgery. The details are in Table 3. The samples were further thawed on ice to subject them to the small extracellular vesicle (sEV) isolation protocol.

### 4.4. IgG Depletion from Mouse/Human Serum Samples

A serum sample of 200 µL was diluted with 50 µL 0.2 µm filtered PBS (Hyclone). Protein A/G plus-agarose beads (Santa Cruz, CA, USA) were added to the sample. The sample was placed on the rotor for 2 h at 4 °C. The sample was centrifuged at 3000× *g* for 10 min and the supernatant was used for sEV isolation. The pelleted beads were further checked for bead-IgG binding to confirm the IgG depletion from the serum samples.

### 4.5. sEV Isolation from Serum

Serum was transferred into the 70 mL polycarbonate ultracentrifuge tube (Beckman Coulter, Brea, CA, USA) and the volume was made up of 0.2 μm filtered PBS. The sample was centrifuged at 37,500× *g* for 1 h at 4 °C in a 45 Ti fixed angle rotor. The sample was filtered through a 0.2 µm syringe filter (Corning, Corning, NY, USA) and spun at 185,000× *g* for 2 h to pellet out sEVs. The sEVs pellet was resuspended in 100 µL PBS. For the Exo Quick^TM^ Ultra kit (System Biosciences, Palo Alto, CA, USA), serum was prior centrifuged at 10,000× *g* for 15 min and sEVs were isolated according to the manufacturer’s protocol. The sEVs from each sample were isolated twice and all the experiments were performed in triplicates. We have submitted all relevant data of our experiments to the EV-TRACK [74] knowledgebase (EV-TRACK ID: EV220431).

### 4.6. Cell Culture Conditions and sEV Isolation from the TNBC Cell Line

MDA-MB-231 triple-negative breast cancer (TNBC) cell line was purchased from Animal Type Culture Collection (ATCC) (Manassa, VA, USA). Cells were grown in the Dulbecco’s Modified Eagle’s Medium (DMEM) (Hyclone, Utah, USA) containing 10% FBS (Gibco, Grands Island, NY, USA) and 1% antibiotic solution (Hyclone). The cell line was incubated at 37 °C under 5% CO_2_. Medium containing 10% FBS was washed twice by adding Phosphate Buffered Saline (PBS) (Hyclone). The minimum amount of medium supplemented with 2% sEV-depleted serum (Gibco, Grand Island) was added to the cells, and cells were grown for 48 h. After 48 h, the medium was collected and subjected to ultracentrifugation in a fixed angle 45 Ti rotor (Beckman Coulter) in a 70 mL polycarbonate bottle. Cells were spun at 2000 g for 10 min to remove cells and cell debris. The supernatant was collected and spun at 37,500 g for 1 h. Media was filtered through a 0.2 μm syringe filter (Corning) and spun at 185,000 g for 2 h to pellet out sEVs. Pelleted sEVs were resuspended in 100 µL 0.2 µm syringe filtered PBS and Nano Tracking Analysis (NTA) was performed.

### 4.7. Nano Tracking Analysis

sEVs numbers and size characterization were determined by measuring the rate of Brownian motion using a NanoSight NS300 system with NTA version 3.4.4 (Malvern Panalytical, Malvern, UK). Samples were diluted 1:100 in 0.2 µm filtered PBS and the relative concentration was calculated according to the dilution factor. The samples were analyzed using camera level, slider gain, and slider shutter set as 14, 366, and 1259, respectively. The detection threshold was kept at 5.

### 4.8. Cryo-Transmission Electron Microscopy

Three to four microliters of the sEVs were added to Lacey carbon grids (300-mesh; Ted Pella, Inc.) that were negatively glow-discharged for 80 s at 30 mA. The excess sample was removed by blotting once for 4 s with filter paper (Ted Pella, Inc.), and then the grid was plunge-frozen in liquid ethane cooled by liquid nitrogen using a Vitrobot plunge-freezer (ThermoFisher Scientific, Waltham, MA, USA). The vitrified samples were imaged using a Glacios 200 kV cryo-transmission electron microscope (ThermoFisher Scientific) equipped with a Falcon 4 camera (ThermoFisher Scientific). The SerialEM software was used to collect images under low-dose conditions at 92,000× magnification corresponding to a pixel size of 1 Å/pixel. For each image, 80 frames were recorded over 8 s exposure time at a dose rate of 8 electrons/pixel/s. The movie frames were aligned using SerialEM.

### 4.9. Sodium Dodecyl Sulphate Polyacrylamide Gel Electrophoresis (SDS-PAGE), Western Blot, and Immunostaining

The whole-cell lysate was prepared in Radioimmunoprecipitation assay buffer (RIPA) lysis buffer with protease and phosphatase inhibitors. For sEV lysates, NTA analysis was carried out for counting sEVs. Volume corresponding to the equal number of sEVs (1 × 10^9^ exosomes) was transferred to the 1.5 mL centrifuge tube and heated in the boiling water bath with RIPA and β-mercaptoethanol. For confirming the depletion of IgG from serum samples before sEV isolation, the pelleted beads bounded with IgG were mixed in RIPA lysis buffer with protease and phosphatase inhibitor and heated in the boiling water bath in reducing conditions with β-mercaptoethanol. The sample was spun at 845 g for 5 min. The supernatant was collected for a further run on SDS-PAGE. All the above-mentioned samples were separated using a 4–12% Bis-tris NuPAGE gel (Life Technologies Corporation, Eugene, OR, USA) and protein was transferred to a nitrocellulose membrane. The membrane was blocked with 5% BSA (Sigma, St. Louis, MO, USA) in TBST solution for 2 h and probed with primary antibodies in 2.5% BSA solution, overnight at 4 °C. The primary antibodies used were as follows: anti-AnxA2 (Rabbit, 1:2000, Cell Signaling, Danvers, MA, USA), anti-Tsg101 (Rabbit, 1:1000, Cell Signaling), anti-Albumin (Rabbit, 1:2000, Cell Signaling), anti-CD9 (Rabbit, 1:2000, Cell signaling), anti-Tsg101 (mouse, 1:1000, BD Biosciences, San Jose, CA, USA), anti-GM-130 (mouse, 1:1000, BD Biosciences), anti-Calnexin (mouse, 1:1000, BD Biosciences), anti-CD81 (Mouse, 1:1000, Santa Cruz Biotechnology, Inc., Dallas, Texas, USA), mouse IgG (mouse, 1:2000, Santa Cruz Biotechnology), and anti-CD9 (Rabbit, 1:1500, Abclonal, Woburn, MA, USA). Further, the membrane was probed with respective HRP-conjugated anti-human IgG, Fc-gamma fragment specific-HRP linked (Human, 1:2000, Cell signaling), secondary goat anti-mouse or anti-rabbit antibody (1:2000, Southern Biotech, Birmingham, AL, USA) in 5% milk in TBST for two hours at room temperature (RT) and was washed three times with TBST. Blots were developed by Immobilon Western Chemiluminescent HRP substrate (Millipore, Burlington, MA, USA) with the Alphaimager machine. The band intensities were calculated by using ImageJ by applying the same conditions to all the blots. The fold change was calculated considering the band intensity of the sample obtained from patient 7 to be 1 as the sample was from stage 1 and had a significant expression of protein for comparison with other samples compared to other three stage 1 sEV samples (Patient 4, 5, and 6).

### 4.10. RNA Isolation, cDNA Synthesis, and Quantitative Reverse Transcription Polymerase Chain Reaction (RT-qPCR)

Total RNA was extracted from the 20 µL of sEV samples by adding 1 mL of TRIzol reagent (Invitrogen, Eugene, OR, USA) to the sample. 200 µL of chloroform (Sigma) was added to the TRIzol reagent and mixed. Samples were vigorously vortexed and incubated for 10 min at RT. Samples were centrifuged at 13,523× *g* for 15 min at 4 °C. The above layer was transferred to another tube with 500 µL of 2-Propanol (Sigma). Samples were mixed by inverting 2–3 times and incubated for 10 min at RT. Samples were centrifuged at 13,523× *g* for 15 min at 4 °C. The pellet was further washed with 75% ethyl alcohol pure 200 proof (Sigma) by centrifuging at 7500× *g* for 5 min at 4 °C. Pellet was air-dried and dissolve in 10 µL of molecular-grade water. RNA quality and concentration were measured by NanoDrop2000 spectrophotometry. 2 µg RNA was used for cDNA synthesis. cDNA was synthesized by using SuperScript™ III First-Strand System (Invitrogen, Eugene, OR, USA). The RT-qPCR was performed by using iQ™ SYBR^®^ Green Supermix (Biorad, Hercules, CA, USA). The Ct values for the mRNA expression were normalized with β-actin. The relative expression of mRNA was calculated using the 2^−ΔΔCt^ method. The primers were purchased from Integrated DNA Technologies. The primer sequences used for human AnxA2 and human β-actin were as follows: AnxA2 forward primer: GCCATCAAGACCAAAGGTGT, AnxA2 reverse primer: TCAGTGCTGATGCAAGTTCC [75], β-actin forward primer: GAGGCTCTCTTCCAGCCTTCCTTCCT, β-actin reverse primer: CCTGCTTGCTGATCCACATCTGCTGG [76].

### 4.11. Statistical Analysis

We carried out the Kruskal-Wallis test to compare the relative fold change in the size and release of sEVs between three or more groups. We applied a Mann-Whitney U test to compare the relative fold changes in the expression of the sEV-derived AnxA2 protein and mRNA between the two groups. The expression was compared between the normal samples and TNBC patient samples from stages 1, 2, and 3 and above. Receiver Operating Characteristic (ROC) curves were constructed to evaluate the efficiency of the biomarkers to predict the response and no response of the neoadjuvant chemotherapy in the patients with high and low expression of AnxA2 protein and mRNA in stages 3 and above derived sEVs. ROC curves were defined by the area under the curve (AUC), their specificities, sensitivities, and accuracies. Simple logistic regression analysis was performed to predict the sEV-derived AnxA2 protein and mRNA as the combined predictive biomarker for the neoadjuvant chemotherapy in TNBC patients with stage 3 and above. A likelihood ratio test result was also considered to effectively use both sEV-derived AnxA2 protein and mRNA to predict the therapy response. Youden’s index was calculated to define the cut-off values of sEV-derived AnxA2 protein and mRNA by using sensitivity and specificity values. Results were expressed as mean ± Standard error mean (SEM). Value of *p* ≤ 0.05 was considered statistically significant as determined by each statistical test where * < 0.05 and ** < 0.01. Statistical analysis was performed using GraphPad Prism 7.04 software.

## 5. Conclusions

The TNBC patient-derived sEVs have increased sEVs-derived AnxA2 protein and mRNA expression in stages 3 and above. The concordance between protein and mRNA in the patient sEVs is very high, hence, we proposed to study the combination of sEVs derived AnxA2 protein and mRNA as the biomarker for chemotherapy response. Our study supports that the AnxA2 protein and mRNA present in the sEVs could be used as a combined biomarker for predicting the responsiveness of neoadjuvant chemotherapy in stages 3 and above in TNBC patients. However, taking into account our small sample size, we propose to investigate a large sample size that could give better sensitivity, specificity, and accuracy to the predictive model for chemotherapy response in TNBC. Our findings need more validation and confirmation with more TNBC patient samples.

## Figures and Tables

**Figure 1 cancers-15-00212-f001:**
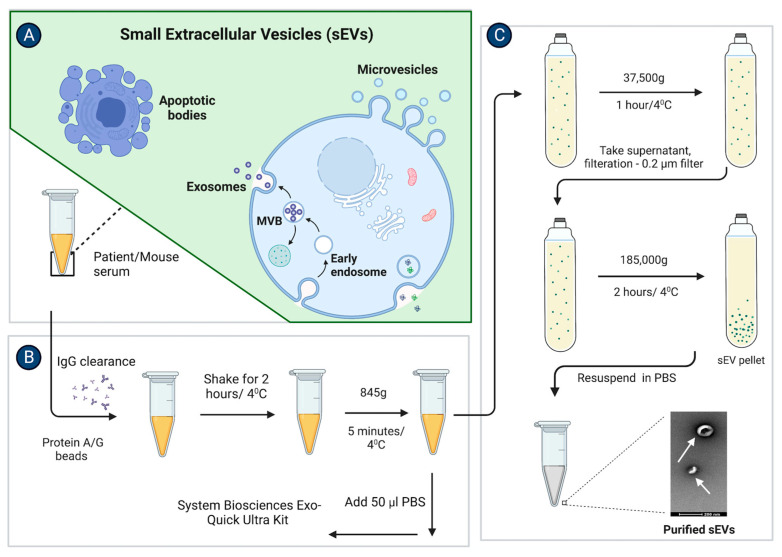
sEV isolation from the mouse or patient serum. Stepwise schematic illustration of the sEVs isolation protocol from a mouse and patient sera as mentioned in (**A**–**C**). (**A**) After collecting the serum from the blood, the serum was subjected to IgG clearance. (**B**) Briefly, samples were diluted to 250 μL by 0.2 μm filtered PBS. 10 μL of Protein A/G plus agarose beads were added to the 250 μL sample. Samples were incubated on the rotor for 2 h and centrifuged at 845× *g* for 5 min at 4 °C. Further samples were subjected to the ultracentrifugation or ExoQuick Ultra kit protocol for sEV isolation. (**C**) For ultracentrifugation, the sample was mixed with 70 mL of 0.2 μm filtered PBS and was spun at 37,500× *g* for 1 h, and at 185,000× *g* for 2 h at 4 °C. The pellet was then resuspended in 100 μL 0.2 μm filtered PBS. For sEV isolation with the ExoQuick Ultra kit, the sample was first diluted with 50 μL of 0.2 μm filtered PBS and was spun at 10,000× *g* for 15 min. Further supernatant was subjected to an ExoQuick Ultra kit protocol. (Adapted from “Extracellular Vesicle Separation by Density Gradient Ultracentrifugation”, by BioRender.com (2020). [Retrieved from https://app.biorender.com/biorender-templates (accessed on 20 December 2022)].

**Figure 2 cancers-15-00212-f002:**
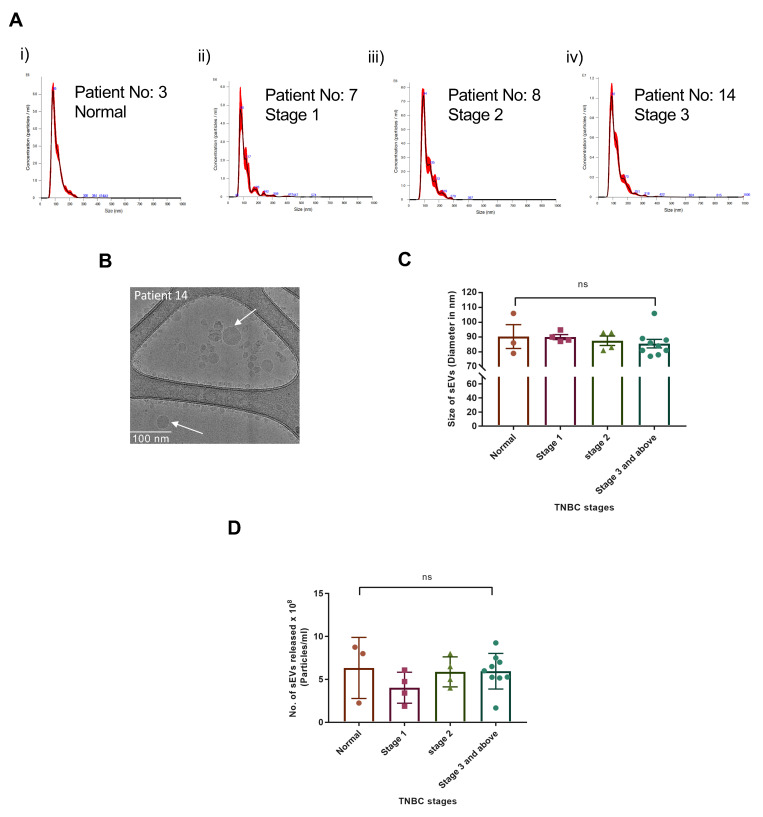
No significant change in sEV sizes and release in patient serum samples of different TNBC stages. (**A**). NTA analysis of Normal and TNBC stage 1, 2, 3 and above patient serum-derived sEV samples isolated by ExoQuick Ultra kit after IgG depletion. (**B**). Cryo-TEM image of TNBC patient serum-derived sEVs. (**C**). Graphical representation shows the comparison of the means of sEV sizes isolated from the patient of each TNBC stage. (**D**). Graphical representation shows the comparison of the means of the number of sEVs released per ml in the 3 normal and 17 patient sera from each stage. (Mean ± SEM, n = 3 for normal samples, n = 4 for stage 1, n = 4 for stage 2, and n = 9 for stage 3 and above).

**Figure 4 cancers-15-00212-f004:**
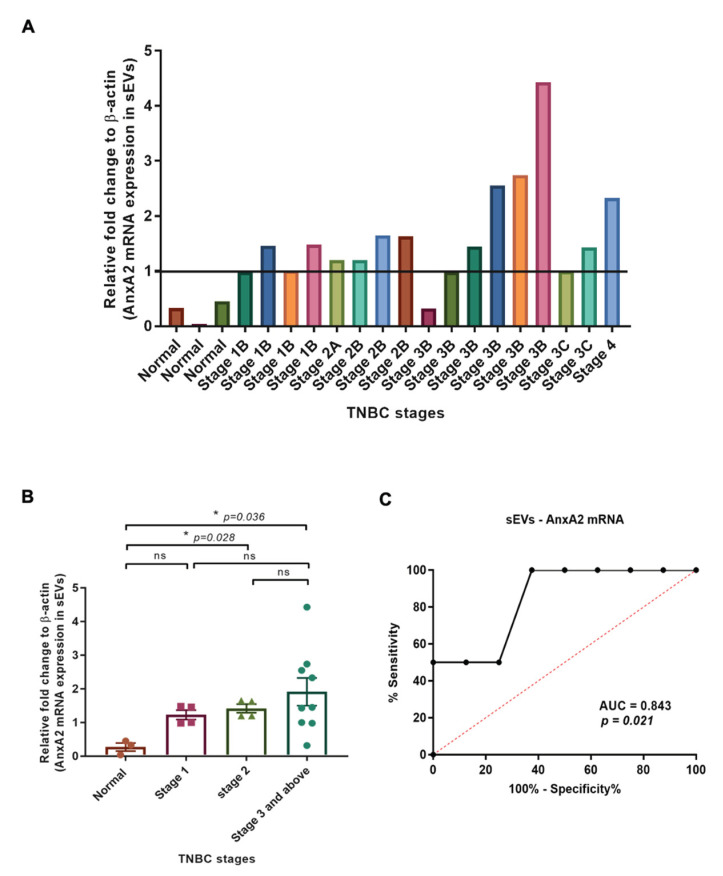
Expression of AnxA2 mRNA in sEVs derived from the TNBC patient serum increases at later stages of cancer and is associated with effective chemotherapy response. (**A**) Analysis of AnxA2 mRNA levels by qRT-PCR in normal and TNBC patient serum-derived sEVs as given in Table 3 from 1–20. The graph shows quantification (fold change) after calculating the 2^−ΔΔCt^ values. The Ct values were taken as the average of two readings for the calculation of 2^−ΔΔCt^ (normal samples—1,2,3; the patient numbers with stage 1-no chemotherapy-4,5,6,7; stage 2-no chemotherapy-8,10, and 11; stage 2 with chemotherapy—9; stage 3 and above-no chemotherapy-14; stage 3 and above with chemotherapy-12,13,15,16,17,18,19,20). (**B**) The graphical representation shows the comparison of the means of the mRNA expression (n-fold change) in sEVs derived from the TNBC patient sample from each stage. (mean ± SEM, n = 3 for normal samples, n = 4 for stage 1, n = 4 for stage 2, and n = 9 for stage 3 and above; * *p* < 0.05). (**C**) ROC curve for serum sEV-AnxA2 mRNA as a biomarker alone for the prediction of responsiveness or non-responsiveness to neoadjuvant chemotherapy in the TNBC patients with stage 3 and above (n = 8).

**Figure 5 cancers-15-00212-f005:**
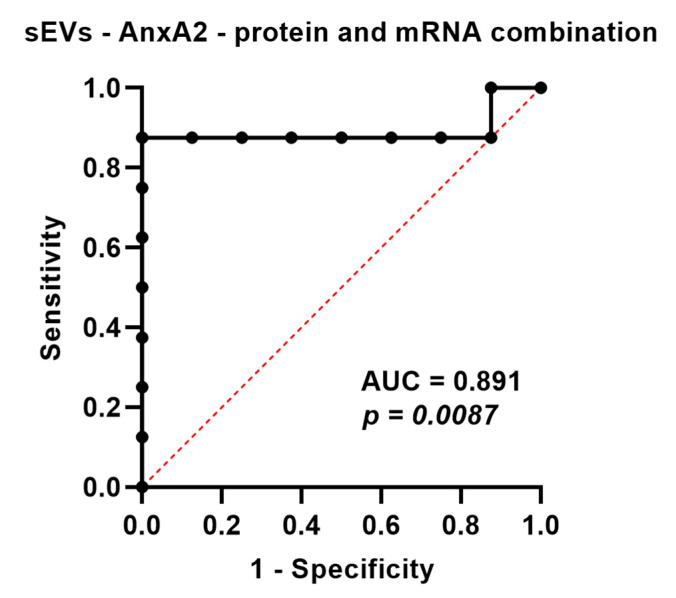
AnxA2 protein and mRNA present in sEVs as a combined biomarker for the prediction of effective neoadjuvant chemotherapy. ROC curve for serum sEV-AnxA2 protein and mRNA as a combined biomarker for the prediction of responsiveness or non-responsiveness to neoadjuvant chemotherapy in the TNBC patients with stage 3 and above (n = 8).

**Table 3 cancers-15-00212-t003:** Patients’ chemotherapy chemotherapy or surgery information and time of sample collection: 3 normal serum samples and 17 TNBC patient samples with information about their clinical prognostic stages, duration of blood drawn—before or after surgery, and the period they underwent chemotherapy.

Patient No.	Clinical Prognostic Stages	Lab Drawn Comparison to Definitive Surgery (in Days)	Was Chemo Given Prior to Surgery? (Yes/No)
1	Non-cancer	-	-
2	Non-cancer	-	-
3	Non-cancer	-	-
4	1B	−5	No
5	1B	−30	No
6	1B	No data	No data
7	1B	−12	No
8	2A	586	No
9	2B	−29	Yes (4 months)
10	2B	−35	No
11	2B	−15	No
12	3B	0	Yes (4.5 months)
13	3B	−7	Yes (4.5 months)
14	3B	477	No
15	3B	234	Yes (4 months)
16	3B	−11	Yes (4 months)
17	3B	−54	Yes (5.5 months)
18	3C	+31	Yes (5.7 months)
19	3C	−20	Yes (4.6 month)
20	4	−12	Yes (6 months)

## Data Availability

The raw data about the conclusions of this article will be made available by the authors, without undue reservation.

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
