# Peer review of "Combination of Small Extracellular Vesicle-Derived Annexin A2 Protein and mRNA as a Potential Predictive Biomarker for Chemotherapy Responsiveness in Aggressive Triple-Negative Breast Cancer"

_cancers, 2022, doi:10.3390/cancers15010212_

Round 1

Reviewer 1 Report

The manuscript " Combination of Small extracellular vesicle-derived Annexin A2 protein and mRNA as a potential predictive biomarker for 3 chemotherapy responsiveness in aggressive Triple-Negative 4 Breast Cancers" is a timely and needed contribution to the field. The examination of circulatory sEVs Annexin is a significant contribution towards exosomal cargoes as biomarkers in TNBC. This study determines the potential usage of circulatory sEVs-derived Annexin protein and mRNA cargoes isolated from the serum of TNBC patients of different stages as a predictive biomarker for responsiveness to chemotherapy in aggressive TNBC. The subject is currently something of a “hot topic” and is one to which the author has made significant contributions. The present study is an appreciable attempt in the respective field. Although the manuscript calls for several edits. The manuscript can be improved by addressing the following suggestions:

1)    It was difficult to follow the author, as there are abrupt transitions in the manuscript that require the author’s attention. I would strongly advise the author to rewrite their introduction, analysis, and discussion to produce a more contextualized introduction to sEVs derived Annexin A2 protein and mRNA as a potential predictive biomarker in TNBC. The paper would benefit from stylistic changes to the way it has been written for a stronger, clearer, and more compelling argument. 

2)    The sample size of each cohort should be stated in the abstract.

3)    It would be useful to describe (briefly) the exact context of use (COU) that this putative biomarker may serve for this small population and when extrapolated in larger n.

4)    It would be good to mention the lack of longitudinal examinations as a limitation to the current study. It is possible that baseline, or longitudinal change, in these markers may offer a different picture as well as expanded potential COUs (e.g., treatment response, predicting progression).

5)    Lines 61-62 Kindly define the term first then use abbreviations; for example, clusters of differentiation (CD9). There are several such errors in the draft, kindly check, and work on it.

6)    In lines 71-73 you introduced bEVs in comparison to sEVs; I could not understand why the author made that comparison in the introduction, as in later parts of the manuscript there was no reference to connect the dots between sEVs and bEVs based on the results shown in the present study. 

7)    Lines 104-122 require re-arrangement as well as fluidity for better reading.

8)    The legend for Figure 1 needs clarification for readers. The figure is good. But as the author has used 1), 2) and 3) in the figure, tits better to avoid confusion and it is advisable to change them to a), b) and c) and re-write the legend by introducing stepwise schematic illustrations in a, b and c.

9)    Line 148 referring “we selected 17 patient 148 samples from a set of over 80 samples and calculated their clinical prognostic stage”.- Why was that done? It’s not clear so kindly explain.

10) Add legends right above/below of each supplemental figure for ease to understand the figures.

11) No doubt the author has followed the MISEV-2018 guidelines for most aspects of sEVs isolation and characterization. But kindly add cryoEM for sEVs from other patients too.

12) Also, as per the guidelines of ISEV; all researchers are strongly encouraged to submit their experimental protocols on EV isolation and characterization to the EV-TRACK website (evtrack.org). The database helps to calculate the metric value for isolated EVs. When the author receives an EV-TRACK ID; kindly add it to the methods in your draft.

13) It was extremely difficult to decipher section 2.2; of the draft, I had to assume and connect the dots a lot, so kindly re-arrange this major piece of your evidence. For presenting the figures of different patients, I suggest creating a group like non-malignant stage 1 no chemo (standard sample patient-7); malignant stage 2 (patient A; B;C); malignant stage 3 (patient X;Y;Z) etc and add it to all the figure legend as required. 

14) Can the author say how many particles were used to isolate total RNA? Did they quantify the RNA: particle ratio? If not, it’s suggested to add this data to their study.

15) Was RNA quantification for mRNA and miRNA conducted? If so, what was the basis of the normalization of RNA amongst treatment and control groups before “reverse transcription of exosomal RNA”?

16) There is limited to no information regarding blood collection, processing, and storage methods. As this is a blood-based biomarker project, a section should be included for the reader.

17) Lines 580-581 referring “ then fold 580 change was calculated considering the band intensity of stage 1 sample (Patient 7) to be 1”. Can the author explain how the band intensity of patient 7 was evaluated before using it as a control for fold change calculation? Kindly add it for readers.

18) There is an interesting finding in this research. However, there is insufficient discussion of exactly what these findings convey to other researchers and their future implications. Unfortunately, reiterating previous research does not suffice the discussion.

19) The author needs to be extremely vigilant of his/her references; in many instances, the references are missing from the required statements. Check carefully and add accordingly.

Author Response

Dear Reviewer,

Hope you are doing well.

Please see the attachment. We highly appreciate you putting your time and efforts into improving our manuscripts.

Best,

Dr. J.K.Vishwanatha

Reviewer 2 Report

In the present study, authors have identified the potential significance of sEV-derived AnxA2 protein and RNA as a predictive biomarker in aggressive TNBC. There are few novel findings and sEV-derived AnxA2 protein and RNA may also hold promising translational relevance for the prognosis of TNBC patients. However, I do notice many important concerns that should be addressed to further enhance the manuscript quality:

1. Why did the authors choose an equal number of sEV rather than total protein concentration for protein expression? They should also load an equal concentration of total proteins present in the sEV. 

2. They should also normalize the protein expression of AnxA2 with respect to some endogenous EV markers such as flotillin-1.

3.  Integrated proteomics and transcriptomics analysis of sEV would be a better approach to identifying novel biomarkers. Why did the authors directly choose AnxA2 in this study rather than performing an Integrated Omics approach?

4. Please also include the expression of AnxA2 in different subtypes of breast cancer including TNBC.

5. It would be great if they could provide the result of AnxA2 expression for overall and recurrence-free survival of breast cancer patients.

6. It would be interesting if they could provide some results with lymph node and distant organ metastasis to establish the functional significance of this marker for predicting metastasis in TNBC patients.

Author Response

Dear Reviewer,

Hope you are doing well.

Please see the attachment. We highly appreciate your efforts and time in improving our manuscript.

Best regards,

Dr.J.K.Vishwanatha

Round 2

Reviewer 1 Report

The draft has been edited properly and I suggest the MS draft be accepted as it is with minor language edits.